# SELF-QUESTIONING LANGUAGE MODELS

## ABSTRACT

**Can large language models improve without external data – by generating their own questions and answers?** We hypothesize that a pre-trained language model can improve its reasoning skills *given only a single prompt* specifying the topic (e.g., algebra word problems) and asking the model to generate its own questions. To do this, we propose Self-Questioning Language Models (SQLM): an asymmetric self-play framework where a proposer is given the topic and generates a question for a solver, who tries to answer it. Both the proposer and solver are trained via reinforcement learning. The proposer receives a reward if the problem is not too easy or too difficult, and the solver receives a reward based on majority voting, a proxy for correctness in the absence of ground-truth answers. For coding, the proposer can instead generate unit tests which are used for verification. We study this asymmetric self-play framework on three benchmarks: three-digit multiplication, algebra problems from the OMEGA benchmark, and programming problems from Codeforces. By continually generating more interesting problems and attempting to solve them, language models can improve on downstream benchmarks without access to any curated training datasets.

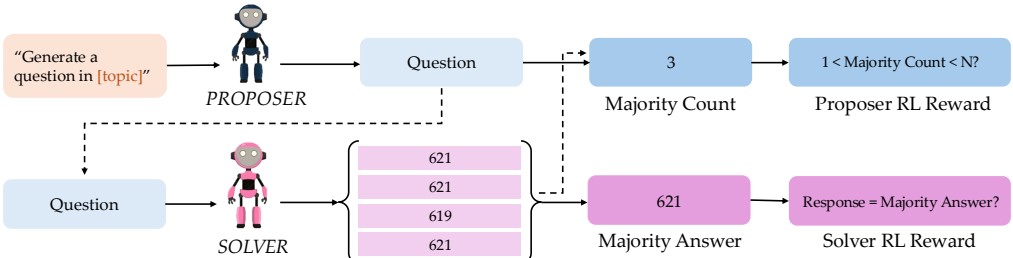

Figure 1: Overview of Self-Questioning Language Models. The only input to the system is a single prompt, given to the proposer. The proposer generates a question related to the given topic, and the solver aims to solve the question. The solver's reward is computed by using the majority vote as a proxy for the ground-truth answer. The proposer's reward is computed based on how many of the answers match the majority answer, to encourage problems not to be too easy or too difficult.

## 1 INTRODUCTION

The post-training of large language models still relies heavily on hand-curated datasets (Yu et al., 2025a; Li et al., 2024), demanding substantial engineering effort and human supervision. In an attempt to alleviate this burden, researchers have developed unsupervised reward functions for reinforcement learning, which use proxies such as the model's internal confidence (Zhao et al., 2025b; Prabhudesai et al., 2025) or the majority answer (Zuo et al., 2025; Shafayat et al., 2025) in the absence of ground-truth rewards or answers. However, these methods still presuppose the existence of well-formed input prompts or questions, and the bottleneck shifts from curating labeled answers to curating high-quality questions – a task that remains labor-intensive and not easily automated at scale. This highlights a critical gap in current methodologies: the lack of scalable, self-sustaining pipelines for generating meaningful questions and answers without human intervention.

Interestingly, pre-trained language models themselves represent a largely untapped resource for addressing this challenge. Much like humans, these models can be seen not only as passive recipients of training data but also as active generators of it. As an analogy, scholars in remote or quiet settings sharpen their thinking through self-directed exploration and questioning. Large language models pre-trained on trillions of tokens of text may possess similar capabilities. By leveraging their intrinsic knowledge and reasoning faculties to simulate both the role of question-asker and responder, models could drive their own intellectual development in a closed-loop fashion. This paradigm opens a compelling avenue for self-supervised post-training that could reduce dependence on curated datasets and unlock a new frontier in autonomous language model refinement.

To explore this paradigm, we propose **S**elf-**Q**uestioning **L**anguage **M**odels (SQLM): an asymmetric self-play framework in which a pre-trained language model is prompted with a high-level domain (e.g., "algebra word problems") and learns to improve by generating and solving its own problems. Asymmetric self-play was first proposed by Sukhbaatar et al. (2017) and applied to goal-conditioned robotic manipulation in OpenAI et al. (2021); here, we apply it to language model post-training, where the model plays two roles: a proposer that creates new problems and a solver that attempts to solve them. Both roles are trained via reinforcement learning. When the generator-verifier gap is small (such as in arithmetic, where solution generation and verification are similarly difficult), we use majority voting over multiple solver outputs as a proxy for correctness. When the gap is large (such as in code generation, where verifying via unit tests is easier than writing a correct solution), we instead ask the proposer to generate unit tests and use that as the solver's reward. The proposer is rewarded for generating non-trivial, solvable problems, while the solver is rewarded for correctness under these domain-specific criteria.

We evaluate this framework on three benchmarks: three-digit multiplication [1], algebra problems from the OMEGA (Sun et al., 2025) benchmark, and coding problems from Codeforces[2]. In every case, training begins with a single prompt describing the task, without any example problems or labeled data. As training progresses, both the proposer and solver improve iteratively, guided by reinforcement signals based solely on internal agreement. Our evaluations show that models trained in this self-supervised manner can demonstrate meaningful gains in reasoning ability without access to any curated training datasets. These results point to the promise of self-play as a general mechanism for self-improvement in language models.

## 2 RELATED WORK

### 2.1 REINFORCEMENT LEARNING FOR LANGUAGE MODELS

Reinforcement learning (RL) has played a pivotal role in the alignment and reasoning capabilities of contemporary large language models. Early efforts primarily focused on aligning model outputs with human preferences via Reinforcement Learning from Human Feedback (RLHF), using algorithms such as Proximal Policy Optimization (PPO) (Schulman et al., 2017) and Direct Preference Optimization (DPO) (Rafailov et al., 2023). In structured domains such as mathematics and code generation, RL has expanded beyond preference modeling to leverage the availability of ground-truth solutions. This enables the use of binary reward signals, allowing for more objective fine-tuning. Approaches like STaR (Zelikman et al., 2022) and $ReST^{EM}$ (Singh et al., 2023) exploit this structure by sampling rationales from the model and selectively fine-tuning on those that yield correct final answers. This selection-based training paradigm encourages more reliable intermediate reasoning steps. A parallel line of work has explored self-refinement capabilities in LLMs, training models to identify and correct their own mistakes through recursive or feedback-driven mechanisms (Kumar et al., 2024; Qu et al., 2024). These methods aim to internalize self-correction as a policy, rather than relying solely on external signals. On the algorithmic front, recent work has introduced RL algorithms specifically tailored for chain-of-thought (CoT) (Wei et al., 2022) reasoning. GRPO (Shao et al., 2024), Dr. GRPO (Liu et al., 2025c), and DAPO (Yu et al., 2025a) have been instrumental in achieving state-of-the-art performance on complex reasoning benchmarks. Most notably, models such as DeepSeek-R1 (Guo et al., 2025; Hu et al., 2025) have demonstrated that targeted RL techniques can substantially enhance general reasoning capabilities.

---

[1] https://github.com/Jiayi-Pan/TinyZero
[2] https://huggingface.co/datasets/MatrixStudio/Codeforces-Python-Submissions

## 2.2 Unsupervised Rewards for Reasoning

Ground-truth answers and external verifiers are often costly or impractical to obtain. This has motivated recent work exploring how far one can go without any form of external supervision. Notably, several studies have demonstrated that pre-trained language models are sufficiently well-calibrated to use their own confidence as a training signal. This has been operationalized through entropy minimization (i.e., reverse KL divergence to a uniform distribution) (Prabhudesai et al., 2025), and forward KL divergence (Zhao et al., 2025b). Another unsupervised reward strategy treats the majority prediction among sampled completions as the "correct" answer (Zuo et al., 2025; Shafayat et al., 2025). These approaches rely solely on the model's internal uncertainty, eliminating the need for labeled data or external evaluators. Building on this line of work, our approach goes a step further: not only does it not require ground-truth answers, it does not even require human-authored questions. Instead, the model itself is used to generate questions, making the pipeline fully self-supervised.

## 2.3 Exploration

Exploration is a foundational challenge in reinforcement learning. To address this, many methods encourage agents to discover novel states through intrinsic rewards. One prominent class is based on prediction error, where novelty is measured by the agent's surprise at its own predictions, such as inverse dynamics models (ICM) (Pathak et al., 2017) or randomly initialized networks (RND) (Burda et al., 2018). Other techniques optimize state entropy to promote diverse state visitation (Liu & Abbeel, 2021). Go-Explore (Ecoffet et al., 2019) separates exploration and robustification, enabling agents to return to promising states and expand from there, while Plan2Explore (Sekar et al., 2020) uses a world model to target states with high epistemic uncertainty. Asymmetric self-play (Sukhbaatar et al., 2017; OpenAI et al., 2021) frames exploration as a game between agents. More recently, exploration concepts have been applied to LLMs for personalization (Wan et al., 2025) and safety alignment (Liu et al., 2025a). Furthermore, recent work (Zweiger et al., 2025) introduces a reinforcement learning loop where models generate their own finetuning data and weight update directives, enabling persistent self-directed adaptation to new tasks and knowledge.

## 2.4 Synthetic Data Generation

As models grow larger and more data-hungry, generating high-quality training data has become increasingly important. Synthetic data generation (Nadas et al., 2025; Wang et al., 2024; Long et al., 2024) has emerged as a powerful strategy for scaling up LLM training. A strong "teacher" model is used to produce high-quality examples, which are then used to train a target model. Notably, synthetic data has played a central role in the development of large-scale models such as K2 (Liu et al., 2025b). In these cases, synthetic data is generated prior to training and treated as a static offline dataset. Fang et al. (2025) proposed an online iterative self-play approach to bootstrap from limited initial data, but in our work, we aim to use no initial data at all. The closest approach to ours is Zhao et al. (2025a), which also focuses on a setting with no initial data; our method is more general and extends beyond verifiable domains such as coding, via the use of majority voting.

## 3 Preliminaries

### 3.1 Reinforcement Learning for Language Models

The language modeling task of generating answers for a given question can be treated as a one-step reinforcement learning problem. For a question $x$ sampled from the dataset $\mathcal{D} = \{(x, y)\}$, the policy $\pi(y_{\text{pred}}|x)$ generates a response $y_{pred}$, for which it receives some reward $r = \mathcal{R}(x, y_{pred}, y)$. The reward is traditionally computed via a measure of similarity of $y_{\text{pred}}$ and $y$, e.g., using hard-coded parsers, model-based verifiers (Ma et al., 2025), or probability-based rewards (Zhou et al., 2025; Yu et al., 2025b). In these settings, RL is used to train the policy to maximize the expected reward

$$\mathbb{E}[r] = \mathbb{E}_{y_{\text{pred}} \sim \pi(y_{\text{pred}}|x)}[\mathcal{R}(x, y_{\text{pred}}, y)].$$

Recently, several unsupervised reward functions $\mathcal{R}(x, y_{\text{pred}})$ have been proposed. These rewards do not rely on having access to the ground truth $y$, instead using majority voting (Zuo et al., 2025; Shafayat et al., 2025), entropy (reverse KL divergence) (Prabhudesai et al., 2025), forward KL divergence (Zhao et al., 2025b), or even random rewards (Shao et al., 2025).

## 3.2 ASYMMETRIC SELF-PLAY

Asymmetric self-play (OpenAI et al., 2021), first proposed for goal-conditioned robotic manipulation, is a method for self-supervised exploration which naturally produces a curriculum of interesting tasks for agents to use for learning. It trains two RL agents: a proposer $P$ who aims to propose challenging tasks, and a solver $S$ who aims to solve those tasks. The proposer receives a reward if the solver fails to solve the task, and the solver receives a reward for solving its assigned task. We refer the reader to OpenAI et al. (2021) for more details on the original application of asymmetric self-play to robotics.

In our language modeling setting, we consider a proposer policy $\pi_{P_t}(x)$ and solver policy $\pi_S(y_{\text{pred}} \mid x)$, where $P_t$ is simply a proposer constrained to a specific topic $t$ (e.g., arithmetic), $x$ is a generated question, and $y_{\text{pred}}$ is an attempt at solving the question. Here, both $P$ and $S$ are language models and trained via reinforcement learning as described in the next section.

## 4 METHOD

### 4.1 MINIMAX OBJECTIVE

The proposer policy $\pi_{P_t}(x)$ and the solver policy $\pi_S(y_{\text{pred}} \mid x)$ are both trained via reinforcement learning to maximize their expected rewards:

$$\text{Solver:} \quad \mathbb{E}_{x \sim \pi_{P_t}, \, y_{\text{pred}} \sim \pi_S(\cdot|x)} \left[ \mathcal{R}_S(x, y_{\text{pred}}) \right],$$

$$\text{Proposer:} \quad \mathbb{E}_{x \sim \pi_{P_t}, \, y_{\text{pred}} \sim \pi_S(\cdot|x)} \left[ \mathcal{R}_P(x, y_{\text{pred}}) \right],$$

This setup involves self-play as the proposer's output is used to condition the solver, and the solver's output is used to compute a reward, which is then used to train the proposer. Figure 1 shows an overview of our method. In the next subsection, we discuss how to design reward functions for the proposer and solver, given that this setting is fully self-supervised in the absence of ground-truth answers and perfect verifiers.

### 4.2 REWARD FUNCTIONS

In the context of self-play without access to ground-truth answers, a central challenge is how to perform verification. One straightforward approach is to have the proposer not only generate a problem but also provide a corresponding solution. The solver's output can then be compared against the proposer's solution to assess correctness. However, there is no inherent reason to trust the proposer's solution over the solver's. In most self-play setups, both models are initialized from the same pre-trained language model (and in our experiments, also share weights). As a result, both are equally susceptible to error, making verification by mutual agreement unreliable.

This issue relates closely to the concept of the generator-verifier gap (Song et al., 2024), which refers to the difference in difficulty between generating a correct solution and verifying the correctness of a given solution. In arithmetic, computing the sum of two numbers and verifying that sum are nearly equivalent in difficulty. However, writing a correct implementation for a programming problem may be difficult, whereas verifying its behavior (e.g., through unit tests) is often easier. Depending on the type of problem, we propose different approaches to designing the proposer and solver.

**When the generator-verifier gap is small.** In domains such as arithmetic, there is no easy way to verify solutions without performing reasoning that is similarly complex to what is required to generate the solution itself. For these cases, we do not ask the proposer to generate a solution, and instead use majority voting (Zuo et al., 2025; Shafayat et al., 2025) as a self-supervised reward for the solver. Specifically, for each problem, we sample $N$ generations from the model and use the majority answer as a proxy for the correct answer. All generations that match the majority answer are given a reward of 1 and all other generations are given a reward of 0.

Let $x \sim \pi_{P_t}$ be a problem sampled from the proposer, and $y_1, \ldots, y_N \sim \pi_S(\cdot \mid x)$ be $N$ independent solver generations. Let $y_{\text{maj}}$ be the majority answer among the $N$ generations. The solver reward is:

$$\mathcal{R}_S(x, y_i) = \begin{cases} 1 & \text{if } y_i = y_{\text{maj}}, \\ 0 & \text{otherwise.} \end{cases} \tag{1}$$

The proposer reward is based on how "reasonable" the problem is: it receives a reward of 0 if all $N$ generations match (too easy) or if none match (too hard), and a reward of 1 otherwise:

$$\mathcal{R}_P(x) = \begin{cases} 1 & \text{if } 0 < |\{y_i : y_i = y_{\text{maj}}\}| < N, \\ 0 & \text{otherwise.} \end{cases} \tag{2}$$

**When the generator-verifier gap is large.** In domains such as coding where verification is easier than generation, we can do better by leveraging the proposer's ability to generate a solution or information useful for verification such as test cases. In this section, we focus on the use-case of coding for concreteness, but one could design variants of this approach for other domains.

Let $x \sim \pi_{P_t}$ be a problem and associated test cases generated by the proposer. Let $y_{\text{pred}} \sim \pi_S(\cdot \mid x)$ be the solver's solution, and let $\text{Tests}(x)$ denote the set of unit tests provided with $x$. Define $\text{Pass}(y_{\text{pred}}, \text{Tests}(x)) \in [0, 1]$ as the fraction of unit tests passed by the solver's output. Then:

$$\mathcal{R}_S(x, y_{\text{pred}}) = \text{Pass}(y_{\text{pred}}, \text{Tests}(x)) \tag{3}$$

The proposer is again rewarded for generating problems that are non-trivial but solvable. It receives a reward of 1 if the solver passes some, but not all, test cases:

$$\mathcal{R}_P(x, y_{\text{pred}}) = \begin{cases} 1 & \text{if } 0 < \text{Pass}(y_{\text{pred}}, \text{Tests}(x)) < 1, \\ 0 & \text{otherwise.} \end{cases} \tag{4}$$

## 5 EXPERIMENTS

### 5.1 SETUPS

**Arithmetic.** We prompt the proposer to generate a three-digit arithmetic problem and use the proposed problem as the input to the solver. The solver is trained via majority voting reward (Zuo et al., 2025; Shafayat et al., 2025) to reinforce the majority answer. The proposer reward is 1 if the problem is not too difficult (the majority answer only appears once, which means all the samples are different) and not too easy (the majority answer is equal to the number of samples $N$, which means all the samples were the same), and 0 otherwise. To evaluate the arithmetic ability of the model, we procedurally generate a test set of 4096 three-digit multiplication problems following the setup from TinyZero [3]. We run our experiments with Qwen2.5-3B-Instruct.

**Algebra.** We prompt the proposer to generate algebra word problems that involve linear equations with up to two variables. The solver and proposer rewards are the same as in the Arithmetic setup. For evaluation, we use the 100-question linear equation test set from the OMEGA benchmark (Sun et al., 2025). We run our experiments with Qwen2.5-3B-Instruct.

**Coding.** We prompt the proposer to generate programming problems similar in style and difficulty to LeetCode easy problems that take in a list of integers and output either a single integer or another list. Unlike in the Arithmetic and Algebra setups, the proposer is also asked to generate five unit tests, and the solver reward is the fraction of unit tests that it passes (0, 0.2, 0.4, 0.6, 0.8, or 1). The proposer reward is 1 if the solver reward is not either 0 or 1, which would indicate that the problem is either too easy or too difficult. To evaluate the coding ability of the model, we evaluate on a subset of the Codeforces[4] test set. containing 123 examples (specifically, the ones from the Eurus-2 (Cui et al., 2025) dataset). We run our experiments with Qwen2.5-Coder-3B-Instruct.

### 5.2 MAIN RESULTS

Table 1 shows results on the respective test sets for Arithmetic, Algebra, and Coding. Without *any external data*, we can improve the accuracy of Qwen2.5-3B-Instruct by $14\%$ on Arithmetic and $16\%$ on Algebra, of Qwen2.5-Coder-3B-Instruct by $7\%$ on Coding. Furthermore, we also compare to a format reward baseline, which gives a reward of 1 to the solver if the format is correct and 0 otherwise (i.e., only teaches correct formatting), instead of the majority voting reward. The results show a significant performance improvement over this baseline, which indicates that these are genuine

---

[3]https://github.com/Jiayi-Pan/TinyZero
[4]https://huggingface.co/datasets/MatrixStudio/Codeforces-Python-Submissions

gains in reasoning ability and the model is not merely learning to format the answer correctly. The format reward is pertinent to Arithmetic and Algebra since the model is expected to put its final numerical answer in a specific format, but is not relevant for coding since the expected solution is long-form text. We show additional results with Llama models in the Appendix.

Table 1: Test set accuracy for Qwen2.5-3B-Instruct with and without self-play and comparison to the format reward baseline. The Qwen2.5-Coder-3B-Instruct model is used for coding. The best results are indicated in bold and the results show standard deviations across three training runs.

|                                | **Multiplication** | **Lin. Eqns.** | **Codeforces** |
| --- | --- | --- | --- |
| *Qwen2.5-(Coder)-3B-Instruct* | 0.791 | 0.440 | 0.320 |
| + self-play | **0.948 ± 0.009** | **0.600 ± 0.010** | **0.391 ± 0.019** |
| + self-play (format reward) | 0.826 ± 0.079 | 0.553 ± 0.015 | N/A |

## 5.3 Qualitative Samples

In this subsection, we study the qualitative behavior of the model. Table 2 shows samples from the Arithmetic task. At step 0, the model outputs a simple arithmetic problem $563 + 247 - 189$, but due to the proposer update, it learns to gradually increase the difficulty of its problems. At step 10, it generates $673 - 145 + 98 \times 2 \div 7$, and at step 20, it generates $384 \div (52 \times 2) + 7^3 - 111$. These arithmetic problems have more terms and involve more combinations of arithmetic expressions, so the model must improve its reasoning abilities in order to solve them. Although the test set consists only of three-digit multiplication problems, training on a wide diversity of self-generated problems is indeed beneficial. See Table 3 for samples from the Coding task. As training progresses, the model learns to generate more interesting and complex problems. In the beginning, it generates very easy problems which can be solved in one line, such as squaring each element of a list. At step 20, it learns to generate an interesting problem of finding the longest contiguous subarray with unique elements. Although we instructed the model to generate problems where the input is a list, one could imagine that this approach could generalize to a wider range of data structures and algorithms.

## 5.4 Proposer Update Frequency

A key hyperparameter in our approach is how often to update the proposer. If it is never updated or updated infrequently, then the model might not learn to generate interesting enough problems. If it is updated too often, the solver might not have enough of a chance to improve on each set of problems and training might be unstable. We study this in Table 4, which shows the performance as we varied the update frequency of the proposer. Overall, we found that updating it every 5 steps worked well across all of our evaluation setups – a good balance between encouraging the proposer to generate more interesting problems and still allowing the solver to make progress. Furthermore, updating the proposer every 5 or 10 steps results in lower variance across runs.

## 5.5 Encouraging Data Diversity

A key consideration when asking the model to generate its own data is how to encourage diversity. In our approach, we generate a single problem at each step using the proposer, and we find that this incremental strategy is sufficient to maintain diversity throughout training. As an alternative, we also explored generating all problems prior to training. Specifically, we generated 16 problems per inference call (since it may be difficult to fit all problems within a reasonable context length) and repeated this procedure 400 times, yielding a dataset of 6,400 questions. We explicitly instructed the model via prompting to ensure that the generated problems spanned a wide range of difficulty, from easy to hard. However, as shown in Figure 2, this pre-generated dataset leads to a noticeable reduction in diversity, ultimately impairing learning on the Arithmetic task. To better understand this phenomenon, we analyzed the generated questions using PCA, which revealed that problems generated online with proposer updates exhibit greater diversity compared to those produced via pre-generation. We hypothesize that models may struggle to operationalize abstract instructions such as to generate "diverse" or "difficult" purely from prompting. Instead, they appear to benefit from

Table 2: Samples from the model over the course of training for the Arithmetic task. The model gradually generates more challenging problems, without making them excessively difficult.

| **(Step 0) Proposer** | $563 + 247 - 189$ |
|---|---|
| **(Step 0) Solver** | `<think>` Let's break down the problem step by step: 
 First, add 563 and 247: - $563 + 247 = 810$ 
 Next, subtract 189 from the result: - $810 - 189$ 
 To perform $810 - 189$, I will subtract the numbers step by step: 
 - Subtract the units place: $0 - 9$, so we need to borrow 1 from the tens place. 
 - After borrowing, this becomes $10 - 9 = 1$. 
 - Now, the tens place value reduces by 1 (from 1 to 0), and we bring down the next digit (8). Since $0 - 8$ is not possible, we borrow 1 from the hundreds place. 
 - After borrowing, this becomes $10 - 8 = 2$. 
 - Now the hundreds place reduces by 1 (from 8 to 7), and we subtract: $7 - 1 = 6$. 
 Putting it together, we get $810 - 189 = 621$. `</think>` 
 `<answer>` 621 `</answer>` |
| **(Step 10) Proposer** | $673 - 145 + 98 \times 2 \div 7$ |
| **(Step 10) Solver** | Let's solve the arithmetic problem step by step. 
 `<think>` First, we need to follow the order of operations (PEMDAS/BODMAS rules), which stands for Parentheses/Brackets, Exponents/Orders, Multiplication and Division from left to right, Addition and Subtraction from left to right. 
 The expression is $673 - 145 + 98 \times 2 \div 7$. 
 1. Calculating the Multiplication and Division: 
 - First, perform $98 \times 2 = 196$ 
 - Then, $196 \div 7 = 28$ 
 Now the expression simplifies to: $673 - 145 + 28$ 
 2. Perform Addition and Subtraction (left to right): 
 - $673 - 145 = 528$ 
 - $528 + 28 = 556$ `</think>` 
 `<answer>` 556 `</answer>` |
| **(Step 20) Proposer** | $384 \div (52 \times 2) + 7^3 - 111$ |
| **(Step 20) Solver** | `<think>` Let's solve the expression step by step: 
 1. Evaluate the parentheses: $52 \times 2 = 104$ 
 2. Now the expression becomes: $384 \div 104 + 7^3 - 111$ 
 3. Perform the division: $384 \div 104 \approx 3.69$; taking integer part gives 3 
 4. Now we have: $3 + 343 - 111$ since $7^3 = 343$ 
 5. Perform the addition and subtraction: $3 + 343 = 346$, $346 - 111 = 235$ `</think>` 
 `<answer>` 235 `</answer>` |

a quantitative, adaptive notion of difficulty provided by the asymmetric self-play process, which continuously updates the proposer's problem distribution in response to the solver's capabilities.

## 6 LIMITATIONS AND FUTURE WORK

While our method requires less manual intervention compared to traditional systems, it does not eliminate prompt tuning entirely. In practice, we found that prompt iteration was necessary to constrain the generation space appropriately and to ensure that outputs adhered to our expected formatting. Although we avoided extensive tuning to preserve generality as much as possible, even minimal manual iteration introduces a bottleneck and potential source of bias. In the future, one

Table 3: Samples from the model over the course of training for the Coding task. The model gradually generates more challenging problems, without making them excessively difficult.

| **(Step 0) Proposer** | You are given a list of integers. Write a program that reads the list and returns a new list containing the square of each integer in the original list. |
|---|---|
| **(Step 0) Solver** | ```python
# Read input from the user
input_string = input()
# Split the input string into a list of strings
input_list = input_string.split()
# Convert each string to an integer and store in a list
int_list = list(map(int, input_list))
# Calculate the square of each integer
squared_list = [x**2 for x in int_list]
# Print the squared list as a space-separated string
print(' '.join(map(str, squared_list)))
``` |
| **(Step 10) Proposer** | You are given a list of integers. Write a program that reads the list and returns the sum of all the even numbers in the list. |
| **(Step 10) Solver** | ```python
# Read the input line of integers
input_line = input().split()
# Initialize the sum of even numbers
sum_even = 0
# Iterate over each integer in the input list
for number in input_line:
    # Convert the string number to an integer
    num = int(number)
    # Check if the number is even
    if num % 2 == 0:
        # Add the even number to the sum
        sum_even += num
# Print the sum of even numbers
print(sum_even)
``` |
| **(Step 20) Proposer** | You are given a list of integers. Write a program that reads the list and returns the length of the longest contiguous subarray with all unique elements. |
| **(Step 20) Solver** | ```python
def longest_unique_subarray(nums):
    # Initialize variables to track the maximum length and the
        current start index
    max_length = 0
    current_start = 0
    # Dictionary to store the last seen index of each element
    last_seen = {}
    # Iterate through the list with the current index as end
    for current_end, num in enumerate(nums):
        # If the element is seen and the last seen index is
            greater than or equal to the current start
        if num in last_seen and last_seen[num] >= current_start:
            # Move the start index to one position ahead of the
                last seen index of the current element
            current_start = last_seen[num] + 1
        # Update the last seen index of the current element
        last_seen[num] = current_end
        # Update the maximum length if the current subarray is
            longer
        max_length = max(max_length, current_end - current_start +
            1)
    # Return the length of the longest subarray with unique
        elements
    return max_length
# Read the input and convert it to a list of integers
input_list = list(map(int, input().split()))
# Call the function and print the result
print(longest_unique_subarray(input_list))
``` |

Table 4: Test set accuracy for self-play with various proposer update frequencies. The best results are indicated in bold and the results show standard deviations across three training runs.

| | Multiplication | Lin. Eqns. | Codeforces |
|---|---|---|---|
| *Qwen2.5-(Coder)-3B-Instruct* | 0.791 | 0.440 | 0.320 |
| frequency = 1 | $0.937 \pm 0.019$ | $0.556 \pm 0.051$ | $0.375 \pm 0.050$ |
| frequency = 5 | $0.948 \pm 0.009$ | $\mathbf{0.600 \pm 0.010}$ | $\mathbf{0.391 \pm 0.019}$ |
| frequency = 10 | $\mathbf{0.951 \pm 0.012}$ | $0.546 \pm 0.005$ | $0.324 \pm 0.014$ |
| frequency = $\infty$ (never) | $0.934 \pm 0.025$ | $0.563 \pm 0.023$ | $0.343 \pm 0.022$ |

could automate this process via a system that can autonomously evolve its own prompting strategy. Furthermore, currently there is no safeguard on the model-generated questions to ensure that they are reasonable, safe, relevant, or interesting. A promising avenue of future might work might be to prompt the LLM to filter or score responses by these characteristics. In addition, our method, like all unsupervised approaches, is fundamentally constrained by the absence of ground-truth rewards or perfect verifiers. In the absence of labeled data, the model must rely solely on internal heuristics, such as self-consistency or majority voting, to assess correctness. This introduces a risk of reinforcement of systematic errors: if the model repeatedly converges on an incorrect solution that is internally self-consistent, there is no mechanism for correction without external guidance. One possible direction to address this is to incorporate labeled datasets in a semi-supervised setting.

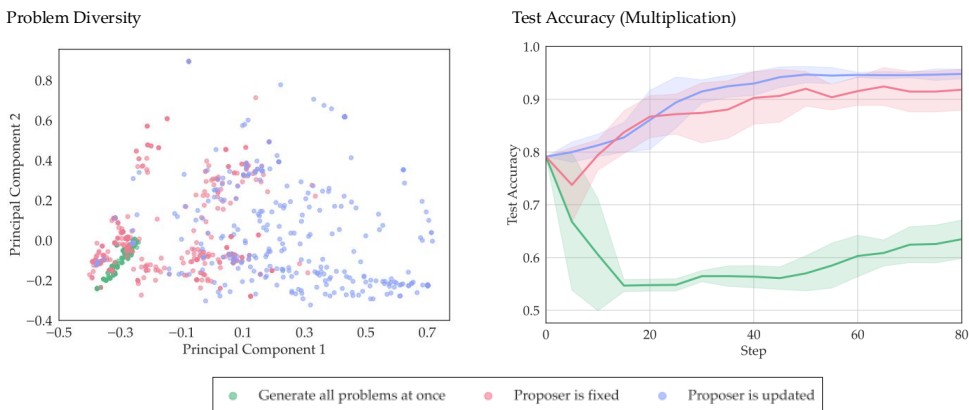

Figure 2: Comparison of generating all problems at once vs one at a time using the proposer.

# 7 CONCLUSION

This paper presents a step toward autonomous language model improvement by introducing Self-Questioning Language Models, a self-improving framework that requires no curated training data. We leverage the intrinsic capabilities of large language models by casting them in dual roles of proposer and solver within an asymmetric self-play setup. By rewarding the generation of problems that are neither too easy nor too difficult, and by reinforcing answers via internal agreement, we demonstrate that models can meaningfully improve their reasoning skills via self-generated content alone. Our experiments across arithmetic, algebra, and code generation tasks show that language models can bootstrap stronger problem-solving capabilities without access to external data. This framework opens a path for fully self-supervised model learning. Our method is not without limitations: prompt design remains a source of hand-engineering, and without external grounding, models can reinforce their own errors. Addressing these issues offers promising directions for future research.

## 8 REPRODUCIBILITY STATEMENT

We described the experimental setup in Section 5.1 and provided a list of hyperparameters in the Appendix. Furthermore, we have submitted the source code as supplementary material.

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

# A  APPENDIX

## A.1  IMPLEMENTATION DETAILS

We build our implementation on top of verl (Sheng et al., 2024). A list of hyperparameters can be found in Table 5. For all experiments, we report the best test accuracy achieved over 100 training steps, averaged over three training runs.

Table 5: Hyperparameters of SQLM.

| Hyperparameter | Value |
|---|---|
| Max prompt length | 512 |
| Max response length | 1024 (Arithmetic) |
| | 3072 (Algebra) |
| | 512 (Coding) |
| Batch size | 64 |
| Policy mini batch size | 32 |
| Policy micro batch size per GPU | 8 |
| Learning rate | $1 \times 10^{-6}$ |
| Weight decay | 0.01 |
| Learning rate warmup | Constant |
| Optimizer | Adam |
| Temperature | 1.0 |
| Top $k$ | -1 |
| Top $p$ | 1 |
| Number of samples per example $n$ | 4 |
| Remove padding | True |
| Use KL loss | True |
| KL loss coefficient | 0.001 |
| Clip ratio | 0.2 |
| Grad clip | 1.0 |
| Proposer update frequency | 5 |

## A.2  MORE RESULTS

We use Qwen2.5-3B-Instruct for our experiments as it is a small model which still has strong math and coding capabilities, as well as good instruction-following capabilities (to enable the proposer to generate reasonable questions). To show the generality of our asymmetric self-play framework, we also show results with Llama-3.2-3B-Instruct and Llama-3.1-8B-Instruct on Codeforces in Table 6.

Table 6: Test set accuracy for Qwen2.5-Coder-3B-Instruct, Llama-3.2-3B-Instruct, and Llama-3.1-8B-Instruct, with and without self-play. The best results are indicated in bold and the results show standard deviations across three training runs.

| | Codeforces |
|---|---|
| *Qwen2.5-Coder-3B-Instruct* | 0.320 |
| + self-play | **0.391 ± 0.019** |
| | |
| *Llama-3.2-3B-Instruct* | 0.211 |
| + self-play | **0.243 ± 0.026** |
| | |
| *Llama-3.1-8B-Instruct* | 0.231 |
| + self-play | **0.382 ± 0.023** |

## A.3 PROMPTS

The prompts used for each task are shown in Table 7. For the Algebra task, we additionally encouraged diversity by asking the model to generate three problems and selecting the last one. For the Coding task, we generated random numbers to show the model the expected format for the test cases. This was done only to make processing of the inputs and outputs easier in the loop, and we intentionally did not give an example problem in order to not bias the proposer.

Table 7: Prompts used for each task.

| | |
|---|---|
| **Arithmetic** | ```"""Generate a three-digit arithmetic problem (up to three digits). Make sure the numbers are not similar and appear unpredictable. Do not solve the problem."""``` |
| **Algebra** | ```"""Create three diverse, challenging algebra word problems that involve linear equations with up to two variables. Use only integers for all coefficients. The problem should be solvable with a unique solution where each variable has a rational (integer or fractional) value. Do not solve the problems. Then select the last one and put it in the format: Selected Question: <question>"""``` |
| **Coding** | ```"""Generate an original programming problem similar in style and difficulty to LeetCode easy problems.``` 

 ```Requirements:``` 
 ```- The problem must take a single line of space-separated integers as input and produce either a single integer or a space-separated list of integers as output.``` 
 ```- Provide 5 test cases in the exact format: INPUT\_STRING ||| OUTPUT\_STRING (no explanations, no extra text, OUTPUT\_STRING must include the trailing '\\n' if needed).``` 

 ```Example Output Format:``` 
 ```Problem Description:``` 
 ```You are given a list of integers. Write a program that reads the list and returns <expected output>.``` 

 ```Input:``` 
 ```A single line contains space-separated integers a\_1, a\_2, ..., a\_n (-1000 <= a\_i <= 1000).``` 

 ```Output:``` 
 ```Print a single integer <expected output>.``` 
 ```Test Cases:``` 
 ```8 -3 7 0 2 ||| 14``` 
 ```-2 5 -4 3 ||| 2``` 
 ```10 -10 ||| 0``` 
 ```4 ||| 4``` 
 ```-5 -1 -4 ||| -10"""``` |

