# OpenReview forum: "Self-Questioning Language Models"
_ICLR.cc/2026/Conference — ICLR 2026 Conference Withdrawn Submission_

### Official Review · Reviewer_2BYK · 2025-10-29

**Soundness:** 3
**Presentation:** 3
**Contribution:** 3
**Rating:** 6
**Confidence:** 3

**Summary:**

This paper proposes a method called Self-Questioning Language Models (SQLM), which aims to enable large language models to improve themselves without any external supervision or labeled data.

The core idea is to introduce an asymmetric self-play framework in which the model takes on two roles:
- Proposer that generates questions or problems, and
- Solver that attempts to answer them

Both agents are optimized through reinforcement learning in an alternating fashion: the proposer is rewarded for generating questions that are neither too easy nor too difficult, while the solver receives rewards based on majority voting or unit-test correctness signals.

Experiments conducted on arithmetic, algebra, and code-generation tasks demonstrate that the model can achieve notable performance gains even without any external data.

Overall, the paper shows that large language models can potentially form a self-improving learning loop—generating, solving, and refining their own data—thus highlighting an interesting direction toward autonomous language model training.

**Strengths:**

- The paper is well-written with clear notations and equations.
- General framework: The proposed approach is conceptually general and, in principle, extendable to various tasks such as reasoning, planning, and code generation.

**Weaknesses:**

- Limited experimental scale: The experiments are conducted only on Qwen 2.5 3B instruct models, lacking results for smaller or larger, or different model families of models.
- Potentially noisy reward signals: The majority-vote reward may reinforce model self-consistency bias, potentially amplifying incorrect consensus among sampled outputs. This could make the model learn inherent biases that are hard to correct later on.
- It seems that the reward is constant for proposer in equation 2 as long as not all the answer or none of the answer is majority. Have authors studied different thresholds?

**Questions:**

- Have the authors considered evaluating the proposed framework on larger-scale models (e.g., 14B or 32B) and across a broader range of LLM families? Currently, experiments are limited to the Qwen and Llama series.
- How stable is the training process between the proposer and solver during self-play? If one agent learns substantially faster than the other, does the system exhibit instability (e.g., oscillations or collapse)?

---

### Official Review · Reviewer_xNos · 2025-10-30

**Soundness:** 2
**Presentation:** 3
**Contribution:** 2
**Rating:** 4
**Confidence:** 3

**Summary:**

This paper introduces Self-Questioning Language Models, a training method in which two LLMs play a generator-discriminator game where a proposer model is rewarded for providing problems of reasonable difficulty, and the solver model is rewarded for solving the problems consistently/correctly. The models are trained using RL on these rewards. Since problems and solution verifiers are all generated/determined by the models being trained, this method works without external training data, and in principle, without verifiable problem solutions. Experiments on 3B scale LLMs show a statistically significant improvement in solve rates on simple arithmetic, linear equation solving, and LeetCode easy problems.

**Strengths:**

The strengths of this paper lie in its interesting perspective on discriminator-generator training for LLMs in order to "bootstrap" reasoning ability.

S1. While the idea of training neural networks with discriminator-generator min-max style games is not novel, applying this to LLM reasoning as an RL objective is a promising approach.

S2. The method is explained clearly and avoids indulging in unnecessary complexity.

S3. The paper demonstrates statistically significant improvement in math and coding, providing a minimum level of significance.

S4. The reward design is especially interesting as it does not rely on any external data or verification algorithms, providing some novelty when compared to mainstream RL techniques for LLM reasoning.

**Weaknesses:**

While the core idea of asymmetric self-play for LLM training without external data is novel, the paper has a mismatch between its claimed contributions and experimental validation. The experiments are conducted on domains where the proposed approach is least needed, and lack sufficient analysis to demonstrate practical significance.

W1. The paper claims to address settings with scarce training data and difficult verification. However, all experiments use arithmetic, algebra, and LeetCode easy problems, domains with abundant training data and straightforward verification. This undermines the core motivation: if verification is easy (arithmetic) or unit tests are available (coding), why not use standard GRPO with existing datasets?

W2. No comparison to simply training on existing curated datasets (e.g., GSM8K for arithmetic, MATH for algebra, existing coding datasets). Without this baseline, it's unclear whether the RL overhead and complexity are justified for the improvements observed

W3. All models tested are small (3-8B parameters). No evidence the approach scales to larger, more capable models. Only one model family primarily tested in main results (Qwen2.5), with limited Llama results in appendix. No experiments on domains where the method would be most useful, i.e, hard-to-verify problems (not in the computational sense, but in the epistemological sense).

W4. Figure 2's PCA visualization lacks details: What is the embedding model? What features are being projected? How is diversity quantified? No systematic analysis of what problem distributions emerge over training.

W5. No investigation of whether models learn general reasoning principles vs. task-specific pattern matching. The acknowledged risk of "reinforcing systematic errors" is not empirically analyzed or addressed. Failure mode analysis and limitations of the majority voting approach would also be nice to see.

**Questions:**

Along with questions raised by the points raised in the 'Weaknesses' section, here are some additional questions.

Q1. During experimentation, did you observe any degenerative cases, where training failed due to biases in the proposer? As mentioned in the paper, these methods are quite sensitive and can suffer from mode collapse. Could the authors provide quantitative data demonstrating the reliability of this approach across various hyperparamters and input prompts?

Q2. A key concern that comes to mind is that, because of the majority voting reward used, there is no way in which the solver LLM can receive any signal on how to correctly solve problems that have common wrong answers. I.e, if it is confidently incorrect, it will never learn to rectify such mistakes. Could the authors comment on how/if this issue is addressed?

Q3. What is the reliability of the majority voting objective? Could authors provide information on the accuracy of the majority voting reward, i.e, how often is the "correct solution" derived from the majority vote actually correct?

Q4. I apologize If I missed this, but it was unclear to me how the solver's solutions are sampled such that they can be considered independent samples. Also, could the authors clarify how many solutions are sampled per problem, and its effect on training behavior?

Thank you for the interesting contribution.

---

### Official Review · Reviewer_YtVC · 2025-11-03

**Soundness:** 3
**Presentation:** 2
**Contribution:** 2
**Rating:** 2
**Confidence:** 3

**Summary:**

The paper looks at improving the performance of an LLM for performing a task by prompting the LLM to generate problems and solutions for the task and using that synthetic data to finetune the model and improve it's performance. It describes basic approaches to verify the problems and solutions generated are good data to finetune on, and then shows the approach works on some different domain types.

**Strengths:**

I agree the approach described of leveraging LLMs to generate synthetic data to finetune on is very promising and successful and should be explored further. The paper describes an implementation of that approach and shows it works on some simple datasets.

**Weaknesses:**

Leveraging LLMs for synthetic data generation to finetune on is a well known and successful approach, and the paper doesn't seem to me to propose anything novel to the technique.

The paper provides very limited experimental results on very simple datasets.

I feel like in 2022 or 2023 this paper would have been considered novel, but in 2025 this approach has been pushed quite a bit farther by many other papers over the last couple years.

**Questions:**

In comparison to the many other synthetic data generation papers out there, does this paper describe any significantly unique or novel approaches?

Does the approach implemented in this paper show SOTA results on datasets that other papers have tried?

---

### Official Review · Reviewer_94Aq · 2025-11-03

**Soundness:** 1
**Presentation:** 2
**Contribution:** 1
**Rating:** 2
**Confidence:** 4

**Summary:**

The paper proposes a self-questioning framework where a small “proposer” model generates problems and a “solver” model answers them and a subsequent  agreement via majority voting serves as an unsupervised reward signal. By intermittently updating the proposer, the system induces a moving curriculum intended to improve reasoning without curated labels. Experiments on lightweight tasks (e.g., three-digit arithmetic, a small linear-equation subset, and a narrow Codeforces slice) show gains over simple baselines, with ablations around proposer update frequency and sample counts. Overall the contributions of the paper need to be better positioned and argued and the supporting experimental design extended.

**Strengths:**

- Timeliness and relevance of the topic. Label-free learning loops for supporting reasoning tasks.

- Overall clear narrative.

**Weaknesses:**

- Narrow experimental design. Lack of clear selection criteria for the target tasks. Limited baselines. Comparisons focus on simple alternatives (e.g., format-reward) rather than strong self-training/self-play methods under matched conditions.

- Lack of a better positioning of the contributions of the work wrt to related work.

**Questions:**

- Can you state more formally the key contributions (articulating against related work)?

- How well do majority-vote and unit-test proxies correlate with expert human judgments of correctness and reasoning quality?

- What mechanisms prevent the proposer from generating flawed, unsafe, or degenerate problems, and how often does this occur in practice?

- Why is the proposer rewarded only for partial agreement, and how do alternative reward designs (e.g., calibrated difficulty, solver uncertainty) affect outcomes?

- How sensitive is performance to proposer update frequency and sampling temperature. Do training curves show oscillations or mode collapse?

- What happens when increasing model size, context length, or moving to harder domains?

- Do gains persist on truly out-of-distribution tasks and adversarially perturbed prompts; how brittle is the solver to noisy or trick questions?

- Why not include stronger self-training/self-play baselines under matched compute and data budgets. How were prompts, sampling, and budgets controlled?

---

### Note · Authors · 2025-12-04

I have read and agree with the venue's withdrawal policy on behalf of myself and my co-authors.